# A Head-to-Head Comparison of ^18^F-Fluorocholine PET/CT and Conventional MRI as Predictors of Outcome in IDH Wild-Type High-Grade Gliomas

**DOI:** 10.3390/jcm11206065

**Published:** 2022-10-14

**Authors:** Ana María Garcia Vicente, Julián Pérez-Beteta, Mariano Amo-Salas, Jesús J. Bosque, Edel Noriega-Álvarez, Ángel María Soriano Castrejon, Víctor M. Pérez-García

**Affiliations:** 1Nuclear Medicine Department, University General Hospital, 13005 Ciudad Real, Spain; 2Mathematical Oncology Laboratory (MOLAB), Department of Mathematics, Faculty of Medicine, Castilla-La Mancha University, 13005 Ciudad Real, Spain; 3Department of Mathematics, Faculty of Medicine, Castilla-La Mancha University, 13005 Ciudad Real, Spain

**Keywords:** ^18^F-Fluorocholine PET/CT, IDH wild-type glioma, multiple glioma, prognosis, SUVpeak to centroid distance, contrast enhanced-MRI

## Abstract

(1) Aim: To study the associations between imaging parameters derived from contrast-enhanced MRI (CE-MRI) and ^18^F-fluorocholine PET/CT and their performance as prognostic predictors in isocitrate dehydrogenase wild-type (IDH-wt) high-grade gliomas. (2) Methods: A prospective, multicenter study (FuMeGA: Functional and Metabolic Glioma Analysis) including patients with baseline CE-MRI and ^18^F-fluorocholine PET/CT and IDH wild-type high-grade gliomas. Clinical variables such as performance status, extent of surgery and adjuvant treatments (Stupp protocol vs others) were obtained and used to discriminate overall survival (OS) and progression-free survival (PFS) as end points. Multilesionality was assessed on the visual analysis of PET/CT and CE-MRI images. After tumor segmentation, standardized uptake value (SUV)-based variables for PET/CT and volume-based and geometrical variables for PET/CT and CE-MRI were calculated. The relationships among imaging techniques variables and their association with prognosis were evaluated using Pearson’s chi-square test and the *t*-test. Receiver operator characteristic, Kaplan–Meier and Cox regression were used for the survival analysis. (3) Results: 54 patients were assessed. The median PFS and OS were 5 and 11 months, respectively. Significant strong relationships between volume-dependent variables obtained from PET/CT and CE-MRI were found (r > 0.750, *p* < 0.05). For OS, significant associations were found with SUVmax, SUVpeak, SUVmean and sphericity (HR: 1.17, *p* = 0.035; HR: 1.24, *p* = 0.042; HR: 1.62, *p* = 0.040 and HR: 0.8, *p* = 0.022, respectively). Among clinical variables, only Stupp protocol and age showed significant associations with OS and PFS. No CE-MRI derived variables showed significant association with prognosis. In multivariate analysis, age (HR: 1.04, *p* = 0.002), Stupp protocol (HR: 2.81, *p* = 0.001), multilesionality (HR: 2.20, *p* = 0.013) and sphericity (HR: 0.79, *p* = 0.027) derived from PET/CT showed independent associations with OS. For PFS, only age (HR: 1.03, *p* = 0.021) and treatment protocol (HR: 2.20, *p* = 0.008) were significant predictors. (4) Conclusions: ^18^F-fluorocholine PET/CT metabolic and radiomic variables were robust prognostic predictors in patients with IDH-wt high-grade gliomas, outperforming CE-MRI derived variables.

## 1. Introduction

Glioblastomas (GBM), the most frequent and aggressive type of primary brain tumor, exhibit significant interpatient differences and a marked intratumoral heterogeneity that results in very different growth patterns and influences the response to therapeutic agents [1].

Gliomas with intact isocitrate dehydrogenase (IDH) defined as IDH-wild type (IDH-wt), the most common astrocytic gliomas, have as characteristics: very high proliferative activity and cellular synthesis, worse prognosis than their mutated (IDH-mut) counterparts and older age at diagnosis than IDH-mut gliomas. They grow into the periventricular white matter adjacent to the subventricular zone and have a higher likelihood of contrast enhancement (CE) on magnetic resonance imaging (MRI), explaining the significant associations of some contrast-enhanced magnetic resonance imaging (CE-MRI) radiomic variables with prognosis [2,3,4,5,6,7].

GBM heterogeneity is notorious and manifests even within the group of IDH-wt tumors. Moreover, molecular data typically available on diagnosis do not allow for capturing the regional heterogeneities present in these tumors [8].

Metabolic imaging using positron emission tomography/computed tomography (PET/CT) offers global tumor information that can be used to assess patient prognosis in glioma. However, because of the limitations of ^18^F-fluorodeoxiglycose, only amino acid PET tracers are recognized as useful tools that provide information complementing MRI in the clinical management of gliomas [9]. Previous works dealing with high-grade glioma patients have been able to distinguish between short and long-term survivors by exploring differences in properties extracted from PET radiomics [10]. However, the experience is limited in the poorest prognosis group, the IDH-wt high-grade gliomas [11].

In the last decade, amino acid PET radiotracers supply constraints in Spain, have limited their use, which has promoted the tentative use of other more accessible PET radiotracers, such as choline analogues. However, although these radiotracers can be useful in the prediction of tumor molecular profiles and prognosis, the experience is limited and not focused on the group of IDH-wt high-grade gliomas [12,13,14]. Thus, the aim of our study was to assess the outcome potential of classical and novel imaging variables (radiomics) obtained from ^18^F-fluorocholine PET/CT and compare their performance with the standard of imaging care CE-MRI together with other clinical factors related to prognosis in patients with IDH-wt high-grade gliomas.

## 2. Materials and Methods

A prospective and multicenter study was designed in 2016, (FuMeGA, Functional and Metabolic Glioma Analysis). Patients with a brain lesion suspicious of glioma after CE-MRI and good performance status (Eastern Cooperative Oncology Group, ECOG ≤ 2) were consecutively included and underwent baseline ^18^F-fluorocholine PET/CT.

The ethics committees of the two participating centers approved the study (internal codes B-176/2016 for Hospital General Universitario de Ciudad Real and 2017/10/104 for the Hospital General Universitario de Albacete.

### 2.1. Patients

Patients with pathologically confirmed IDH-wt high-grade gliomas and at least 15 months of clinical follow-up after ^18^F-fluorocholine PET/CT were selected from our prospective FuMeGA dataset and included for the present analysis.

Informed signed consent was obtained from all patients. Performance status of patients was obtained (before and after surgery).

Patients underwent adjuvant standard treatment, defined by Stupp protocol (complete or not) based on their clinical status. Complete protocol was defined when radiotherapy combined with temozolomide followed by adjuvant temozolomide for at least 6 months was administered. Clinical and imaging follow-up was performed every three months.

Overall survival (OS) was defined as the time, in months, from the date of surgery or biopsy until death or last follow-up examination. Progression-free survival (PFS) was defined as the time, in months, from the date of surgery or biopsy until tumor progression or last follow-up examination. Tumor progression was established by histological confirmation or by imaging/clinical follow-up, using the Response Assessment in Neuro-Oncology (RANO) criteria on the first abnormal CE-MRI [15].

### 2.2. MRI and PET/CT Acquisition and Evaluations

All baseline MRIs were acquired in the axial plane with a 1.5-T MRI unit. Parameter ranges were as follows: repetition time msec/echo time msec, 6–25/3–10. After segmentation, we computed several geometric measures previously reported to correlate with survival outcomes for GBM: total volume, CE volume, necrotic volume and CE rim width. Two surface-related measures were included: (a) the total surface and (b) the surface regularity. 

Patients included satisfied the following criteria: available volumetric pretreatment T1 CE-MRI (section thickness ≤ 1.60 mm, spacing between sections ≤ 1.6 mm, gap ≤ 0 mm, pixel spacing ≤ 1.00 mm), no substantial imaging artifacts and presence of CE areas.

All CE-MRIs were analyzed by the same image expert (J.P., with 9 years of expertise in tumor segmentation) without normalization of the raw gray-level values.

The Digital Imaging and Communications in Medicine files were imported into the scientific software package Matlab (R2021b, MathWorks, Natick, Mass) and were preprocessed using an in-house semiautomatic segmentation procedure. First, a gray-level threshold chosen to identify the largest CE tumoral volume automatically delineated tumors. Then, segments were manually corrected slice by slice. Necrotic tissue was defined as hypointense (i.e., nonenhancing) tumoral regions inside CE tumor tissue.

Multilesionality was defined as the visualization or two or more separate contrast-enhanced lesions on CE-MRI.

A set of 3D volumetric measures was computed: CE volume (CEV), the volume surrounded by the CE areas or inner volume (IV), in most cases the inner volume corresponding to necrotic areas, and the total postcontrast T1w tumor volume (V = VCE + IV). Spherical rim width (δs), a spherical approximation of the average size of the CE rim, was computed as in [7]. Finally, the maximum tumor diameter in 3D (maxD), defined as the maximal distance between two points located on the tumor surface, was also computed.

The extent of resection (partial or complete) or biopsy was established on CE-MRI. Partial resection was defined as a case when any residual tumor was observed on the T1-weighted CE-MRI performed within 48–72 h post-surgery, and complete resection was when no residual tumor was observed.

PET/CT scans were performed in single hybrid equipment (Discovery DSTXL-1, General Electric) located at the reference center following a previously used procedure [14].

Two experienced nuclear medicine physicians (A.M.G.V. and A.S.C.) independently assessed the PET scans in an Advantage Windows station (v.4.). In case of disagreement, a third evaluator revised the images.

In visual assessment, any uptake higher than background was considered positive. Multiple gliomas, defined as multifocal or multicentric, were assessed using PET. Tumors were considered multicentric when two or more separate hypermetabolic foci were observed. If the multiple foci were connected by a faint increased metabolic background uptake, the tumor was considered multifocal. The two subgroups were considered together as multilesional for the statistical analysis.

Tumors were segmented on PET images using a semiautomatic procedure described in a previous study [14]. Morphological variables such as the maximum diameter and tumor total surface were included. Several metabolic parameters were obtained including classical metabolic standardized uptake value (SUV)- and volume-based variables (SUVmax, SUVmean, SUV peak, metabolic tumor volume (MTV) and total lesion activity (TLA)). A variable that informs of the radiotracer intensity distribution, the coefficient of variation (COV), was computed. COV is a statistical measure of the dispersion around the mean of points in a dataset [16]. It was calculated as the standard deviation of SUV/SUVmean following the formula where $\sigma_{SUV}$ is the standard deviation of SUV:$$COV=\frac{\sigma_{SUV}}{SUV_{mean}}$$

Tumors with COV below 0.30 were defined as homogeneous, and those with COV ≥ 0.30 were classified as heterogeneous [16].

Geometrical variables as sphericity and novel geometrical variables, the SUVpeak to centroid distance (SpCD) and the SUVmax to centroid distance (SmCD), were obtained. Sphericity is given by the following equation:$$S_{g}=6\sqrt{\pi }\frac{V}{\sqrt{S^{3}}}$$

It quantifies the irregularity of the shape of the radiotracer uptake in three dimensions [17], where total volume (V) is measured in cubic centimeters and the total surface (S) is measured in square centimeters. SR is a dimensionless ratio between the segmented tumor volume and the volume that a spherical tumor with the same surface would have.

Tumors with very irregular shapes have sphericity values close to 0, while spherical ones will have sphericity values close to 1.

SUVmax to centroid distance (SmCD) was proposed as a measure of tumor evolutionary status since previous research has shown that the further the hotspot of activity from the center of the tumor, the worst the patient prognosis [18]. SpCD was defined as the distance between the cube center used for the SUVpeak calculation and the metabolic centroid or geometrical center of the tumor on ^18^F-fluorocholine PET/CT (in millimeters).

SUVpeak was the maximum distribution of the 3D segmentation of the tumor in which the SUV of every voxel was averaged with its 26 neighbor voxels. To calculate this, to the spatial distribution of SUV in the tumor, we applied an image filter (convolution) with the kernel $\frac{1}{27}J_{3\times 3\times 3}$, i.e., a 3D matrix where all the elements equaled $\frac{1}{27}$.

The location of the SUV peak was denoted as $\left(x_{sp},y_{sp},z_{sp}\right)$. Next, in the same frame of reference, we calculated the centroid of the N segmented voxels (segmentation centroid, SCD), whose coordinates $\left(x_{c},y_{c},z_{c}\right)$ are computed as:(1)$$x_{c}=\frac{1}{N}\overset{N}{\underset{i=1}{\sum }}x_{i},\;y_{c}=\frac{1}{N}\overset{N}{\underset{i=1}{\sum }}y_{i},\;z_{c}=\frac{1}{N}\overset{N}{\underset{i=1}{\sum }}z_{i}$$

The distance between the SUVpeak location and the SCD, was calculated as the Euclidian distance between the two points:(2)$$SCD=\sqrt{{\left(x_{sp}-x_{c}\right)}^{2}+{\left(y_{sp}-y_{c}\right)}^{2}+{\left(z_{sp}-z_{c}\right)}^{2}}$$

It is desirable to normalize this distance by a measure of size to make it independent of the volume of each individual case. For that purpose, we divided the SCD by the mean spherical radius (MSR), which is the radius that a sphere having the same volume as the tumor would have and acts as a linear measure of volume. For a tumor containing *N* segmented (active) voxels and *N*’ necrotic (non-active) voxels, all of them with a voxel volume of *V_v_*, the total volume is $V=\left(N+N\prime \right)\times V_{v}$. Then, the mean spherical radius is computed as $MSR=\sqrt[{3}]{3V/\left(4\pi \right)}.$ This quantity allows us to define the normalized SCD as nSCD=SCD/MSR.

This way we have a relative measure of the SUVpeak location with respect to the tumor center. Values of nSpCD close to 0 correspond to cases where the SUVpeak is located near the center while values approaching 1 will be close to the lesion boundary [18].

In the case of multiple lesions, the largest one was considered in computing these measures.

### 2.3. Molecular Analysis

After samples were processed, immunohistochemistry protocols were completed with Dako-Omnis/Agilent^®^. IDH1-2 mutation status was assessed using polymerase chain reaction according to standardized protocols. Only IDH-wt tumors were included in the present analysis.

Regarding Ki-67, a cut-off value of 20% of positively stained cells was used to separate patients into groups of low and high proliferation according to previous studies [19,20].

### 2.4. Statistical Analysis

Statistical analysis was performed using SPSS software (v.24 IBM, New York, NY, USA). Qualitative variables were summarized using percentages and frequencies, and quantitative variables using the mean and standard deviation (SD).

The analysis of association between clinical (age, ECOG), histopathologic (histology, Ki-67 proliferation index), therapy (extent of surgery and adjuvant treatment), metabolic and morphologic (derived from PET/CT and CE-MRI, respectively) and prognostic variables, which defined the events of interest (death and progression), was performed using Pearson’s chi-square test. The following values of r > 0.75 and *p* < 0.05 were considered robust and statistically significant, respectively.

OS and PFS survival times, discretized with respect to the medians and means of the PET variables for both groups (greater and lower than the median), were compared using *t*-test. The normality of PET variables was checked with the Kolmogorov–Smirnov test.

Receiver operator characteristic (ROC) curves were computed to find cut-offs for PET and MRI variables able to predict the prognosis, using median OS and DFS. A categorical classification was defined for nSpCD using its mean.

Survival analysis was performed using the Kaplan–Meier method and Cox regression analysis to study the relationship between clinical, PET and MRI variables with prognosis. The log-rank Mantel–Cox test was used to compare the survival distributions among the categories of the qualitative variables; *p*-values were considered relevant and statistically significant when *p* < 0.05.

## 3. Results

A total of 54 patients with IDH-wt high-grade gliomas were assessed, 33 (61.1%) men, with a mean age of 61.44 ± 12.64 years. Most of the patients (51, 94.4%) had an ECOG of 0 at the time of tumor diagnosis; 2 (3.7%) had an ECOG of 1, and 1 (1.9%) had an ECOG of 2. Table 1 shows patient and tumor characteristics. Only half of the patients underwent complete resection, and 63% completed the Stupp protocol. In seven patients, it was impossible to analyze CE-MRI due to either problems with the images (either noise due to large voxel size or voxel interspacing or patient movement during the scans or presence of other artifacts) or very diffuse tumors without well-defined boundaries or absence of lesion enhancement on CE-MRI. Multiple brain lesions were found in a third of patients on PET/CT, whereas on MRI, only 6.4% of the patients had multiple lesions.

Most patients progressed (51/54) and died (48/54) during the first 12 months of follow-up. Median PFS and OS were 5 and 11 months, respectively. A group of 14 patients received second-line chemotherapy due to progression.

With regards to imaging variables, the volume-based homonym variables such as total volume, surface and maxD were larger on PET than in MRI. However, sphericity showed similar ranges in both imaging modalities. Mean PET- and CE-MRI-derived variables are shown in Table 2. Significant associations between PET and CE-MRI size dependent variables were found, as larger tumors on CE-MRI were associated with greater metabolic tumor burden (MTV and TLA) on PET. With less intensity of association, albeit with significant results, bigger tumors had a higher SmCD (Table 3).

For the discretized variables of PFS and OS (greater and lower than the median), only the means of the SUV-based variables for OS showed significant differences between groups. Means and SDs for each group and *p*-values are shown in Table 4.

In ROC curve analysis, significant associations with OS were found for SUVmax, SUVpeak and SUVmean. The cut-offs of SUVmax = 3.56, SUVpeak = 2.55 and SUVmean = 1.25 provided AUCs of 0.710 (*p* = 0.009) with sensitivity and specificity of 0.70 and 0.71, respectively; 0.701 (*p* = 0.012) with a sensitivity and specificity of 0.70 and 0.63, respectively; and 0.669 (*p* = 0.034) with sensitivity and specificity of 0.70 and 0.71, respectively for all the SUV-based variables. None of the MRI variables showed significant associations with prognosis.

In 44/54 (81.5%) lesions, the SUVpeak was located further from the centroid than half of the radius. In order to select two homogeneous groups, we used the mean nSpCD (0.66) to separate them.

We found that ^18^F-fluorocholine PET delineated significantly larger volume-based variables than CE MRI in all cases (Table 2). On average, the PET-derived volume was larger than the defined by CE MRI by a factor of 1.64. Figure 1 and Figure 2 show examples of those discrepancies.

Only the selected cut-offs of SUVmax, SUVpeak and SUVmean showed significant association with OS. Log rank curves for OS and PFS are shown in Figure 3 and Figure 4. Metabolic multilesionality was also a predictor of OS (Figure 4).

Using Cox regression analysis, significant associations of OS with SUVmax were found [HR: 1.17, 95% CI = (1.01, 1.35), *p* = 0.035), SUVpeak (HR: 1.24, 95% CI = (1.01, 1.52), *p* = 0.042] and SUVmean [HR: 1.62, 95% CI = (1.02, 2.58), *p* = 0.040]. An inverse relation of OS with sphericity was observed [HR: 0.80, 95% CI = (0.66, 0.97), *p* = 0.022], where the HR is expressed for every tenth of increase of the variable. Multilesionality showed close to significant association with OS [HR: 1.79, 95% CI = (0.98, 3.25), *p* = 0.057]. Among clinical variables, only Stupp protocol (no complete vs complete) and age, showed significant associations [HR: 2.81, 95% CI = (1.51, 5.09), *p* = 0.001 and HR: 1.04, 95% CI = (1.01, 1.07), *p* = 0.005, respectively]. Extent of resection, WHO category and Ki-67 did not show significant association (*p* = 0.147, *p* = 0.808 and *p* = 0.304, respectively).

Age and Stupp protocol showed significant associations with PFS [HR: 1.03, 95% CI = (1.01, 1.05), *p* = 0.024 and HR: 2.16, 95% CI = (1.21, 3.85), *p* = 0.009, respectively]. Extent of resection, WHO category and Ki-67 did not show significant association (*p* = 0.912, *p* = 0.787 and *p* = 0.299, respectively). No PET variable was significantly associated with PFS, although sphericity was close to the significance [HR: 0.83, 95% CI = (0.67, 1.32), *p* = 0.095]. Ki-67 was not associated neither with PFS or OS.

No CE-MRI variable showed significant association with either OS or PFS.

In multivariate analysis using Cox regression, several clinical and metabolic variables such as age, Stupp protocol, multilesionality and sphericity were significant predictors of OS, while for PFS, only age and Stupp protocol showed significant associations (Table 5): The risk of death increased by 4.4% and the risk of progression increased by 2.7% for every year of increase in age. On the other hand, patients who did not receive complete Stupp protocol had greater risk of death and progression than the patients who received the treatment (2.81 and 2.19 times, respectively). With respect to metabolic variables, patients with multilesionality on ^18^F-flurocholine PET/CT had 2.2 times greater risk of death than patients with unifocal lesions. Finally, sphericity was a protective factor decreasing the risk of death by 20% for every tenth of increase in the sphericity (Figure 5 and Figure 6).

## 4. Discussion

Conventional MRI has a key role in anatomical structure definition in brain tumors. Several studies have shown that the combination of CE-MRI with PET/CT could be more sensitive in identifying tumor tissue and peritumoral normal brain tissue, solving the known limitations of biopsy location and/or surgical resection decisions based solely on CE-MRI information [21,22,23,24].

CE areas on T1-Gd MRI correlate with the most active/proliferative tumor regions and correspond to brain areas with nonfunctional blood vessels [25]. Resections are usually planned based on CE area, but nonenhancing diffuse infiltration between the CE part of the tumor and the peritumoral edema, known as the transitional zone, is of great interest; this infiltration may explain the high rate of tumor recurrences, in most cases localized to the resection margin, even with radical macroscopic resections [26,27,28]. In CE lesions, the external boundary of the CE areas is routinely used to define the tumor volume. However, PET volume significantly exceeds tumor volume on CE-MRI, and sometimes even on FLAIR or T2 weighted images [29,30,31]. Song et al. [31] described an improved assessment of tumor extent using ^18^F-fluoroethyl-L-tyrosine (FET) PET as compared with CE-MRI. In the present work, although associations between volume-related variables of ^18^F-fluorocholine PET and CE-MRI were found, tumor volume was also found to be substantially larger on PET than on MRI.

In relation to tumor morphology, an irregular surface was associated with a poor prognosis using CE-MRI [2,17,32]. Pérez-Beteta et al. [17] did not find significant associations between T1-weighted MRI volume and prognosis, with surface regularity and CE rim being the only significant survival predictors in patients with GBM. In the present work, for every tenth of increase in sphericity on ^18^F-fluorocholine PET/CT, the death risk decreased in an 20%.

Using ^18^F-FET, a long survival time in IDH-wt glioma patients was associated with a smaller biological target volume at initial diagnosis [13]. However, the availability of amino acid is limited in some countries, which requires exploring other alternatives such as choline analogues. In the present work, significant associations between SUV-based variables and OS were found.

Multiple GBM, defined as multifocal or multicentric on CE-MRI, can occur in up to 20% of cases. Some of these GBM contact the subventricular zone involving the cortex, which confers a worse prognosis [33]. ^18^F-fluorocholine PET/CT detected multiple lesions in a larger number of patients than CE-MRI (33.3% vs 6.4%, respectively). Although the limited spatial resolution is a limitation of PET imaging, its molecular dependence is a virtue based on the robust association of this variable with prognosis in our sample of patients.

IDH enzymes participate in a variety of metabolic mechanisms that by catalyzing the oxidative decarboxylation of isocitrate play a potential role in oncogenesis, such as the Krebs cycle, glutamine metabolism, lipogenesis, redox regulation, and cellular homeostasis [34]. In fact, the IDH status in gliomas is currently the cornerstone for the characterization of the most aggressive tumors, based on the new WHO classification [35]. In our study, although we used the previous WHO classification mainly based in histopathology analysis, our focus on IDH-wt tumors aligned our analysis of GBM with the new classification. Meanwhile, Ki-67, a DNA-binding nuclear protein expressed throughout the cell cycle only in proliferating cells, was associated with the histological grade, increased volume, and risk of recurrence and/or death in patients with glioma [36,37,38,39], but the utility of Ki-67 in IDH-wt classification has been scarcely investigated. Armocida et al. [19] documented a negative association between Ki-67 and PFS in GBM, showing that a Ki-67 staining percentage above 20% predicted poorer survival in IDH-wt GBM. In our work, Ki-67 had no significant association with the prognosis, probably because most patients had high Ki-67 values. Although the experience is limited, other works have reported more adverse metabolic conditions (higher SUV and MTV) in IDH-wt and high Ki-67 tumors [14,40].

The impact of clinical variables such as preoperative performance status on survival in patients with GBM is well established [41]. Regarding the extent of surgery, most patients with IDH-wt high-grade glioma and no gross total removal of the tumor were observed to experience in-field or out-field failures within 1 year after diagnosis [42]. Moreover, several clinical conditions as patient performance status and others related to the administered treatments, such as extent of resection and adjuvant radio-chemotherapy, have been described as important predictors of OS. However, age of patients has a substantial impact on prognosis [43]. In the present work, Stupp protocol was a robust OS and PFS predictor. Patients who underwent complete Stupp protocol showed better OS and PFS [44]. Extent of resection did not show any significant relationship with prognosis, probably due to a more limited survival benefit in more aggressive GBM compared with others described by other authors [45,46]. Furthermore, most of our patients had good performance status, which limited the possibility of using this variable as a reliable outcome factor. Moreover, our prognostic cut-off using median values of OS and DFS did not fit with the classical survival groups attending to time thresholds of 36 months for long-term and 15 months for short-term survivors. In our cohort, we could not assess these differences mainly due to the reduced number of long-term survivors (only two patients), probably explained by the poor prognosis of these tumors [47,48].

GBM is one of the most heterogeneous tumor types, containing multiple tumor genotypes and phenotypes with different signatures, and its clinical course can be influenced by the variety and proportion of different cellular subtypes within each tumor [49,50]. PET seems a potential tool for the stratification of the prognosis of IDH-wt tumors attending to different imaging characteristics. As an example, even the well-known phenomenon of the interplay between the proliferation and migration of tumor cells in GBM, described as its “go and grow” pattern, could be assessed with metabolic imaging [51]. Niches of cancer stem cells (CSCs), mostly located close to the tumor boundaries, have an unlimited proliferation potential and display finger-like proliferation patterns along the tumor contour [52,53]. The cell diffusion on the tumor border, addressing multiple GBM, could be defined on ^18^F-fluorocholine PET/CT. Multiple GBM, including both multifocal and multicentric tumors, are indicative of a highly malignant and invasive phenotype with worse prognosis than unifocal GBM [33,54]. Based on the lack of significant differences in OS between multifocal and multicentric GBM, we considered the two groups together and assessed this feature on ^18^F-fluorocholine PET/CT [55]. In our opinion, sometimes difficulties exist in the classification of multifocal vs multicentric tumors using metabolic imaging based on the limitations of finding “metabolic connections”: the reduced resolution of current scanners or their irregular morphology reflects the spatial aspects of biological heterogeneity within these tumors.

Additionally, the polyclonality of CSCs may have an impact on both tumor shape and heterogeneity [56]. The COVs revealed an inhomogeneous intensity voxel distribution, with values higher than 0.30, although no significant effect was found in patient prognosis in the present work.

Previous clinical studies and mathematical models support that the greatest proliferation is found near the interface between the solid tumor and the surrounding healthy tissue in GBM [27,57]. On the other hand, when the tumor grows, there is a peripheral displacement of more proliferative components explained by: (i) a decrease in the nutrient concentration at the center due to the lack of effective neovascularization and (ii) space constraints that make the duplication faster in the boundary than in the bulk. This dual phenomenon marks the appearance of a central necrotic core with the peripheral displacement of more proliferative cells seeking nutrients, and it explains the existence of a proliferative rim [58]. Therefore, in more proliferative tumors, the most malignant cells are expected to be located at the tumor border. Accordingly, the malignancy of cells may increase in average along the tumor radius: the further from the center, the more malignant the cell should be. In the present work, bigger tumors demonstrated a higher SmCD, which seems to support the previous hypothesis.

Choline analogue radiotracers are markers of cell membrane synthesis and are thus proliferation surrogates [59,60]. Mathematical models incorporating evolutionary dynamics on PET have shown that peak metabolic activity is expected to increase in magnitude and to move towards the tumor boundary as human solid tumors progress [17]. Similar results were obtained when analyzing tumor specimens concentrating active Ki-67 cells in the external portion of the tumors [61]. A previous work by our group evaluating patients with breast and non-small-cell lung cancers revealed the strength of the novel metric, normalized distance from peak activity (SUVmax) to centroid as a prognostic biomarker in comparison with the classical metrics [17]. In the present work, we used SUVpeak as a more reproducible, stable, and accurate metric than SUVmax [62,63]. However, in our sample of IDH-wt high-grade gliomas, no significant association of nSpCD with prognosis was found for the threshold chosen. The highly advanced proliferation in all GBMs could explain the finding that in most of the analyzed tumors, the SUVpeak was located further than the half of the radius. This fact probably limited the statistical potential of this variable as a marker of prognosis in these intrinsically more aggressive tumors.

Regarding the limitations, the small sample could have affected our results, although we did find significant and relevant associations of some PET radiomics with patient prognosis compared with CE-MRI. In addition, because of the difficulties in the co-registration of hybrid images acquired with different equipment, we did not assess the voxel-level metrics of spatial similarity between PETs and CE-MRIs. However, based on the discrepancies observed between the two volumes, we anticipate a substantial mismatch between CE and PET positive areas.

With respect to the strengths, we developed and explored the prognostic potential of the SCD, a novel geometric variable that might be able to define the geometric location of the peak metabolic activity in the tumor, as a surrogate marker of the invasiveness and proliferative power of IDH-wt high-grade gliomas. Furthermore, ^18^F-fluorocholine PET data could help in classifying patients into different prognostic groups. That information added to the improved discrimination of prognosis associated with the introduction of IDH status in comparison with the old WHO grade classification could help with identifying high-risk patients who warrant more aggressive therapies and closer follow-up.

## 5. Conclusions

Less spherical and multiple lesions on ^18^F-fluorocholine PET/CT, patient age, and Stupp protocol were relevant metabolic and clinical variables associated with the poorest prognosis in patients with IDH-wt high-grade gliomas.

We identified that ^18^F-fluorocholine PET/CT offers additional and robust information, with respect to CE-MRI, for finding the subgroup of IDH-wt glioma patients with the worst prognosis, establishing a basis for molecular subtyping using PET that could be defined in future works.

## Figures and Tables

**Figure 1 jcm-11-06065-f001:**
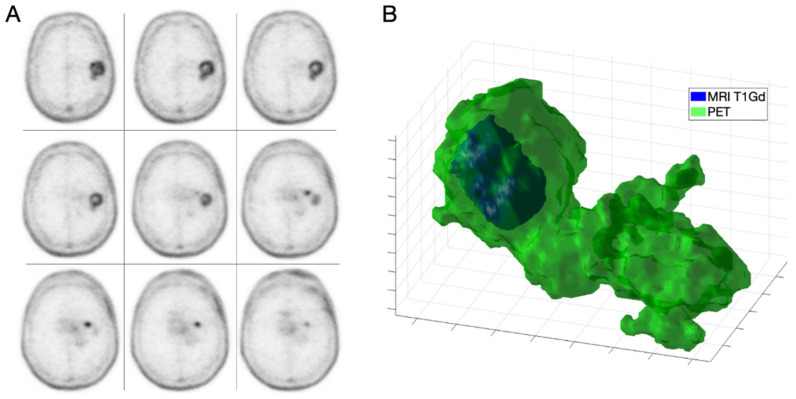
IDH-wt high grade GBM in a 68-year-old male (patient 1). Axial slices of ^18^F-fluorocholine PET (**A**) and combined three-dimensional representation of segmented tumor on MRI T1-Gd and PET (**B**) with total volumes of 3.97 and 36.45 mL, respectively.

**Figure 2 jcm-11-06065-f002:**
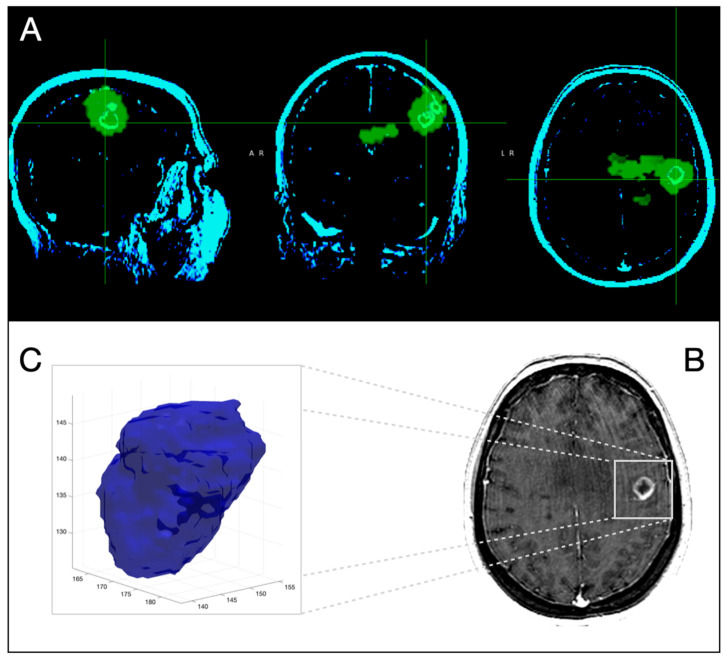
Three-dimensional corregistration of MRI T1-Gd with ^18^F-Fluorocholine PET for the same patient as in Figure 1. Axial, coronal and sagittal slices representative of PET (green) and CE-MRI (light green) (**A**) Representative axial slice of CE-MRI (**B**). Three-dimensional representation of segmented CE-tumor on MRI (**C**).

**Figure 3 jcm-11-06065-f003:**
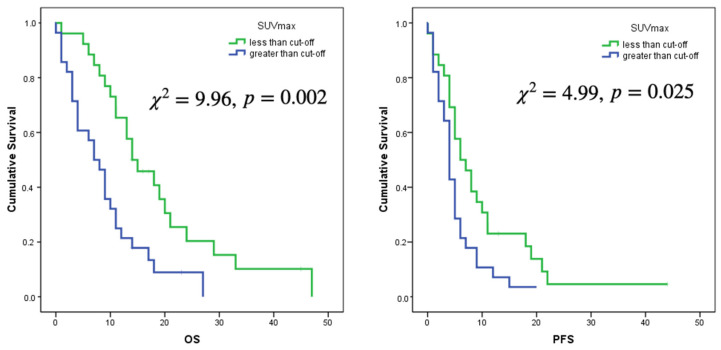
Log rank curves for OS and PFS for a cut-off of SUVmax of 3.56 and a cut-off of SUVmean of 1.25.

**Figure 4 jcm-11-06065-f004:**
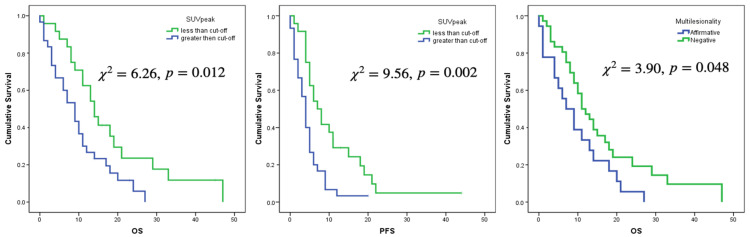
Log Rank curves for OS and PFS for a cut-off value of SUVpeak of 2.5 and for OS regarding multilesionality or not on ^18^F-Fluorocholine PET images.

**Figure 5 jcm-11-06065-f005:**
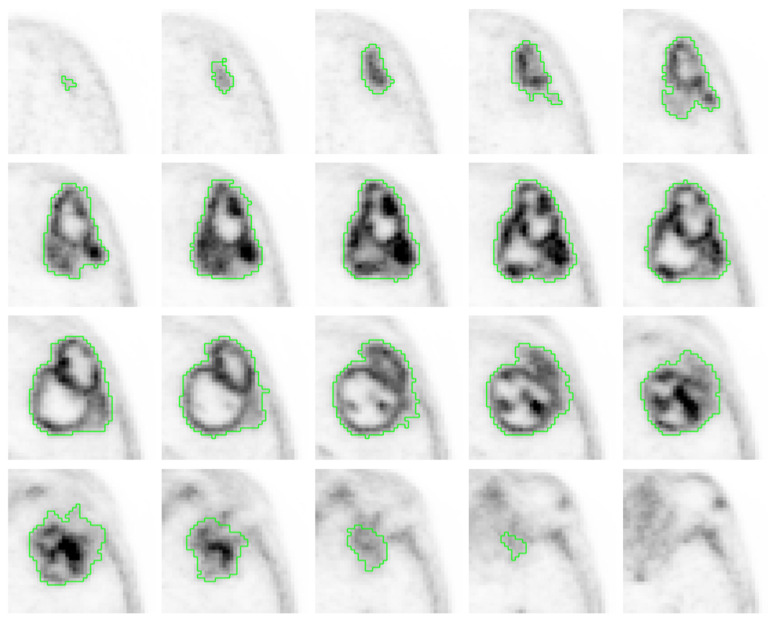
Axial ^18^F-fluorocholine PET slices of a segmented IDH-wt high grade GBM of a 78-year-old male (patient 2) who underwent gross total resection and incomplete Stupp protocol. PFS and OS were 5 and 6 months, respectively. The visual analysis shows a heterogeneous radiotracer distribution (COV of 0.68), with high ^18^F-fluorocholine activity in some peripherical locations (SUVmax of 5.46) with a SmCD of 19 mm.

**Figure 6 jcm-11-06065-f006:**
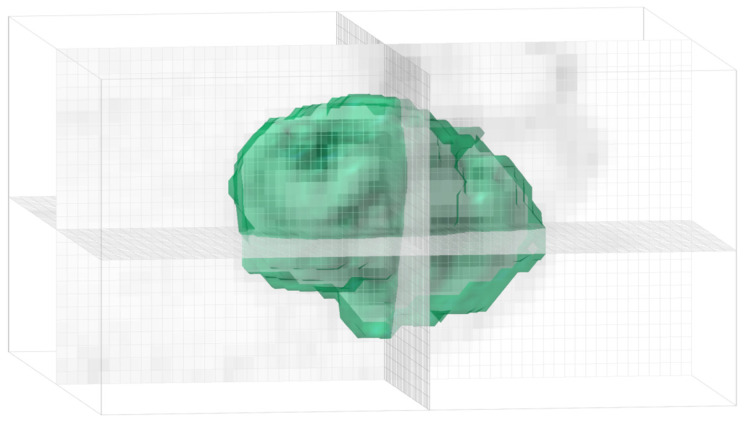
Bubble chain morphology and low sphericity (0.56) in the three-dimensional representation of the segmented tumor of patient 2 (see Figure 5).

**Table 1 jcm-11-06065-t001:** Patients’ tumor characteristics (*n* = 54).

Characteristics	*n* (%)
Histology/WHO category	
Anaplastic astrocytoma/III	6 (11.1)
Glioblastoma/IV	48 (88.9)
Ki-67 category	
High (≥20%)	31 (57.4)
Low (<20%)	12 (22.2)
n.a.	11 (20.4)
Tumor resection	
Complete	27 (50.0)
Partial	13 (24.1)
Biopsy	14 (25.9)
Post-surgery ECOG	
0	46 (85.2)
1	5 (9.3)
2–3	3 (5.5)
Multiple lesions on FCH PET/CT	
Yes	18 (33.3)
No	36 (66.7)
Multiple lesions on CE-MRI (*n* = 47)	
Yes	3 (6.4)
No	44 (93.6)
Treatment (Stupp complete)	
Yes	34 (63.0)
No	20 (37.0)

*n* = number of cases, (%) percentage respect to total cases, SD: standardized deviation, WHO: World Health Organization, n.a.: not available, IDH: isocitrate dehydrogenase, GBM: glioblastoma, CE-MRI: contrast enhanced postcontrast T1-weighted magnetic resonance images, ECOG: Eastern Cooperative Oncology Group, FCH: ^18^F-Fluorocholine.

**Table 2 jcm-11-06065-t002:** Overall mean values of radiomics obtained on ^18^F-fluorocholine PET/CT and T1-Gd MRI.

FCH PET/CT Variables	Mean ± SD
SUVmax	4.04 ± 1.89
SUVmean	1.37 ± 0.58
SUVpeak	3.01 ± 1.35
MTV (mL)	42.68 ± 26.30
TLA (mL)	56.15 ± 42.15
Total volume (mL)	46.14 ± 30.20
COV	0.46 ± 0.12
Surface (s.c.)	87.68 ± 53.27
Sphericity	0.63 ± 0.14
maxD (cm)	6.00 ± 1.76
SmCD	14.87 ± 5.46
SpCD	13.71 ± 5.48
nSpCD	0.66 ± 0.23
**T1-Gd MRI Variables**	**Mean ± SD**
CE volume (mL)	18.53 ± 13.62
Total volume (mL)	28.41 ± 25.28
Surface (s.c.)	67.46 ± 47.43
maxD (cm)	4.99 ± 1.65
Sphericity	0.52 ± 0.11
CE width (cm)	0.62 ± 0.27

FCH: ^18^F-fluorocholine, SD: standard deviation, SUV: standardized uptake value, MTV: metabolic tumor volume, TLA: total lesion activity, COV: coefficient of variation, (s.c.): square centimeters, (mL): milliliters, (cm): centimeters, SmCD: SUVmax to centroid distance, SpCD: SUVpeak to centroid distance, nSpCD: normalized SUVpeak to centroid distance; maxD: maximum diameter; CE: contrast enhanced.

**Table 3 jcm-11-06065-t003:** Relation between ^18^F-Fluorocholine PET/CT and T1-Gadolinium derived MRI variables.

PET/CT Variables	CE-MRI Variables					
	Total Volume	CE Volume	CE Width	maxD	Surface	Sphericity
SUVmax	r = −0.210	r = 0.085	r = 0.198	r = 0.128	r = 0.002	r = −0.016
	*p* = 0.890	*p* = 0.570	*p* = 0.181	*p* = 0.391	*p* = 0.989	*p* = 0.914
SUVmean	r = −0.150	r = −0.088	r = 0.082	r = 0.016	r = −0.126	r = −0.037
	*p* = 0.314	*p* = 0.557	*p* = 0.582	*p* = 0.917	*p* = 0.399	*p* = 0.805
MTV	r = 0.776	r = 0.779	r = 0.212	r = 0.679	r = 0.794	r = −0.290
	*p* < 0.001	*p* < 0.001	*p* = 0.152	*p* < 0.001	*p* < 0.001	*p* = 0.048
TLA	r = 0.491	r = 0.573	r = 0.330	r = 0.549	r = 0.513	r = −0.230
	*p* < 0.001	*p* < 0.001	*p* = 0.024	*p* < 0.001	*p* < 0.001	*p* = 0.119
COV	r = −0.030	r = 0.079	r = 0.316	r = −0.027	r = −0.024	r = 0.071
	*p* = 0.842	*p* = 0.598	*p* = 0.030	*p* = 0.857	*p* = 0.874	*p* = 0.634
maxD	r = 0.430	r = 0.443	r = 0.109	r = 0.416	r = 0.475	r = −0.182
	*p* = 0.003	*p* = 0.002	*p* = 0.466	*p* = 0.004	*p* = 0.001	*p* = 0.222
Surface	r = 0.751	r = 0.701	r = 0.115	r = 0.614	r = 0.780	r = −0.280
	*p* < 0.001	*p* < 0.001	*p* = 0.441	*p* < 0.001	*p* < 0.001	*p* = 0.056
Sphericity	r = −0.480	r = −0.381	r = 0.044	r = −0.304	r = −0.470	r = 0.146
	*p* = 0.001	*p* = 0.008	*p* = 0.769	*p* = 0.038	*p* = 0.001	*p* = 0.328
SmCD	r = 0.408	r = 0.410	r = −0.027	r = 0.522	r = 0.510	r = −0.327
	*p* = 0.004	*p* = 0.004	*p* = 0.855	*p* < 0.001	*p* < 0.001	*p* = 0.025
nSpCD	r = −0.067	r = −0.059	r = −0.157	r = 0.084	r = 0.040	r = −0.118
	*p* = 0.653	*p* = 0.694	*p* = 0.293	*p* = 0.574	*p* = 0.789	*p* = 0.429

CE-MRI: contrast-enhanced magnetic resonance imaging, maxD: maximum diameter, MTV: metabolic tumor volume, TLA: total lesion activity, COV: coefficient of variation, SmCD: SUVmax to centroid distance, nSpCD: normalized SUVpeak to centroid distance, r: Pearson’s correlation coefficient, *p*: *p* value.

**Table 4 jcm-11-06065-t004:** Differences in the means of metabolic variables in patients with higher and lower or equal median PFS and OS.

	PFS≤ 5 Months, *n* = 31 (Mean ± SD)	PFS > 5 Months, *n* = 23 (Mean ± SD)	*p* Value
SUVmax	4.40 ± 1.96	3.56 ± 1.72	0.107
SUVmean	1.48 ± 0.64	1.21 ± 0.45	0.093
SUVpeak	3.25 ± 1.34	2.69 ± 1.31	0.129
MTV (mL)	42.60 ± 25.83	42.80 ± 27.51	0.978
TLA (mL)	61.83 ± 45.82	48.49 ± 36.20	0.254
COV	0.46 ± 0.12	0.46 ± 0.13	0.919
Surface (s.c.)	88.52 ± 53.14	86.55 ± 54.62	0.895
Sphericity	0.61 ± 0.14	0.65 ± 0.12	0.229
maxD (cm)	6.09 ± 1.80	5.89 ± 1.73	0.696
SmCD	15.03 ± 5.54	14.65 ± 5.48	0.800
nSpCD	0.66 ± 0.25	0.67 ± 0.21	0.868
	**OS≤ 11 Months, *n* = 30 (Mean ± SD)**	**OS > 11 Months, *n* = 24 (Mean ± SD)**	***p* Value**
SUVmax	4.57 ± 2.02	3.37 ± 1.50	0.019
SUVmean	1.51 ± 0.66	1.18 ± 0.38	0.032
SUVpeak	3.39 ± 1.42	2.55 ± 1.11	0.021
MTV (mL)	42.59 ± 28.11	42.80 ± 24.46	0.978
TLA (mL)	62.20 ± 47.91	48.59 ± 33.05	0.242
COV	0.46 ± 0.13	0.46 ± 0.11	0.990
Surface (s.c.)	88.42 ± 55.78	86.76 ± 51.14	0.910
Sphericity	0.60 ± 0.14	0.66 ± 0.13	0.117
maxD (cm)	6.10 ± 1.80	5.89 ± 1.74	0.663
SmCD	15.55 ± 5.99	14.02 ± 4.71	0.310
nSpCD	0.69 ± 0.23	0.62 ± 0.23	0.231

PFS: progression-free survival, OS: overall survival, SD: standard deviation, SUV: standardized uptake value, MTV: metabolic tumor volume, SmCD: SUVmax to centroid distance, nSpCD: normalized SUVpeak to centroid distance, TLA: total lesion activity, COV: coefficient of variation, (s.c.): square centimeters, (mL): milliliters, (cm): centimeters, maxD: maximum diameter, *n* = number of cases.

**Table 5 jcm-11-06065-t005:** Multivariate analysis of clinical and ^18^F-fluorocholine PET/CT radiomics variables with OS and DFS (*).

	OS			PFS		
	*p*-Value	HR	95% CI	*p*-Value	HR	95% CI
Age	0.002	1.044	1.016–1.073	0.021	1.027	1.044–1.051
Stupp complete (No vs Yes)	0.001	2.813	1.519–5.209	0.008	2.197	1.228–3.930
Multiple lesions (Yes vs No)	0.013	2.203	1.177–4.122			
Sphericity	0.027	0.788	0.637–0.973			

OS: overall survival, PFS: progression-free survival, CI: confidence interval, HR: hazard ratio, (*) only significant associations are shown.

## Data Availability

All data generated or analyzed during this study are included in this published article.

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
