# Peer review of "A Head-to-Head Comparison of 18F-Fluorocholine PET/CT and Conventional MRI as Predictors of Outcome in IDH Wild-Type High-Grade Gliomas"

_jcm, 2022, doi:10.3390/jcm11206065_

Round 1

Reviewer 1 Report

Thank you for the opportunity to review the article titled:

Head-to-head comparison of 18F-Fluorocholine PET/CT and 2 conventional MRI as predictors of outcome in IDH wild-type 3 high grade gliomas.

General comments:

The article is interesting and based on the original database. Thus, it deserves attention. The authors used various parameters to support the presented hypotheses and conclusions.

Specific comments:

Abstract:

The content is difficult to follow, too many elements make the content imprecise and not really telling. Please, consider rephrasing, removing sentences in brackets, focus on the importance of the study by answering specific questions: why do we conduct the research? What tools have we used to achieve our objectives? What main, beneficial results have we obtained and what leading conclusion have those results provided?

Introduction:

One of many objectives of the scientific manuscript is to provide multiple background data that help all the Readers to understand the need to conduct new research. Brain tumour diagnosis is very difficult and complex. Therefore, especially in the nuclear medicine field, providing more data in the Introduction section seems to be of value. There are multiple methods and radiopharmaceuticals exercised in CNS evaluation. That includes not only specific types of tumours but also metastatic lesions. It seems to be valuable to provide a bit more data (not only for the general interest but also to exceed the citeability of the article). It would also be valuable to provide guidelines regarding brain tumour imaging. Additionally, the Introduction lacks insights regarding the value of the research. The manuscript is valuable and interesting and the Authors should enhance the importance of their hard work.

Please, check:

https://link.springer.com/article/10.1007/s00259-014-2961-x

https://doi.org/10.3390/ph14080722

https://doi.org/10.3390/ijms20194669

I would kindly advise limiting the number of text indentations which suggests starting a new thought. These are most often continuous paragraphs that do not demand new lines.

Methods:

Please, provide ethics at the beginning of the section.

Please, consider reordering the “Patients” section: who, why, and how plus clearly specified inclusion/exclusion criteria.

Please, be concise: all the patients data in the “Patients” section, and protocols in the dedicated section.

Statistical analysis – I appreciate providing more data than just general statements. Please, notice that ROC curves are usually performed to find specific cut-off values.

Results:

Please, consider providing Patients’ epidemiologic data in Methods instead of the Results table.

Moreover, the therapeutic protocol should also be mentioned in the Methods (patients’ characteristics). Additionally, the reason for choosing specific parameters should be provided and explained in detail in the Methods section + discussed in the Discussion section. In the Results, the Authors could enhance the importance of the tumour characteristics mentioned (maybe also background data in the Introduction?). In Table 1, the Authors mixed: patients’ characteristics, tumour characteristics, etc. It should be separated.

The Authors did not comment on any tumour data after providing those. Please, clarify. Why do Authors think it is important to present those? It is meaningful but it demands comment.

Tables 2,3,4 - a similar situation to Table 1. There are multiple data with no comment. Please, explain Radiomics in the Introduction section, provide appropriate data in Methods, comment in Results, and discuss in Discussion. The article does not speak for itself, and it definitely should. After reading it, the audience should know more, not less. The content is highly up-to-date and very interesting. The data obtained by the Authors are credible and very time-consuming to obtain and describe. Please, enhance that.

Please, consider adding Youden Index to ROC curves. It is very important to show the strength of the ROC evaluation.

Please, provide the Table with statistical observations. Please, make sure the statistical significance of the results is always mentioned. 

Table 5 – above-mentioned comments regarding the Tables are applicable in this condition as well.

Discussion:

Please, consider the comments made above. Make sure that all the elements raised by the Authors are discussed. Enhance the importance and utilities of the research.

Conclusions:

Please, make sure you presented, explained and discussed all of the data supporting conclusions. Please, enhance the importance of the data referring to the conclusions as they are very briefly mentioned and not highlighted enough to draw those.

Reviewer 2 Report

The authors present an interesting study on the use of PET-CT in the study of gliomas. Such a relatively low-cost radiological tool allows an analysis of metabolic activity that is easy to understand and also useful for prognostic purposes. I believe it improves our understanding of the pathology and to be at most accurate in prognostic framing. I only recommend that we especially consider in the introduction that with the new WHO 2021 classification, glioblastoma is in fact by definition included in the wild type form and therefore reformulate the study from this point of view.

Round 2

Reviewer 1 Report

Thank you for adjusting the content. The article has been significantly improved.